# Targeting the Complement Cascade for Treatment of Dry Age-Related Macular Degeneration

**DOI:** 10.3390/biomedicines10081884

**Published:** 2022-08-04

**Authors:** Prem N. Patel, Parth A. Patel, Matthew R. Land, Ibrahim Bakerkhatib-Taha, Harris Ahmed, Veeral Sheth

**Affiliations:** 1Department of Ophthalmology, University of Texas Southwestern Medical Center, Dallas, TX 75390, USA; 2Department of Ophthalmology, Medical College of Georgia, Augusta University, Augusta, GA 30912, USA; 3University Retina and Macula Associates, Oak Forest, IL 60452, USA; 4Department of Ophthalmology, Loma Linda University Medical Center, Loma Linda, CA 92350, USA

**Keywords:** age-related macular degeneration, geographic atrophy, complement cascade, therapeutics, clinical trials

## Abstract

Age-related macular degeneration (AMD) is the leading cause of irreversible vision loss in the elderly population. AMD is characterized in its late form by neovascularization (wet type) or geographic atrophy of the retinal pigment epithelium cell layer (dry type). Regarding the latter type, there is growing evidence supporting an association between the pathophysiology of dry AMD and key proteins in the complement cascade. The complement cascade works as a central part of the innate immune system by defending against foreign pathogens and modified self-tissues. Through three distinct pathways, a series of plasma and membrane-associated serum proteins are activated upon identification of a foreign entity. Several of these proteins have been implicated in the development and progression of dry AMD. Potential therapeutic targets include C1q, C3, C5, complement factors (B, D, H, I), membrane attack complex, and properdin. In this review, we provide an understanding of the role of the complement system in dry AMD and discuss the emerging therapies in early phase clinical trials. The tentative hope is that these drugs may offer the potential to intervene at earlier stages in dry AMD pathogenesis, thereby preventing progression to late disease.

## 1. Introduction

Age-related macular degeneration (AMD) is a debilitating condition that represents one of the leading causes of visual impairment among the elderly population in developed countries [1]. Estimates currently place the global affected population between 30 and 50 million individuals, a figure expected to rise to almost 300 million by 2040 [2]. As a consequence of its prevalence and the associated progressive and inexorable central vision loss, the societal burden of this disease is immense [3].

Classification of AMD is established on clinical observation. Although multiple systems have been developed [4,5,6], the predominant scale defines disease progression into three stages according to features such as drusen size and absence or presence of pigmentary changes, geographic atrophy (GA), and neovascularization. Early and intermediate stage disease is characterized by variation in the presence and size of drusen (>63 µm) and potential retinal pigment epithelium (RPE) abnormalities. Late stage disease is characterized by elements of substantial advancement, including GA and choroidal neovascularization (CNV) [7]. GA is the definitive outcome in the progression of nonexudative (dry) AMD, whereby degeneration of macular photoreceptors and RPE cells irreversibly atrophies retinal tissue. Alternatively, CNV is the hallmark of exudative (wet) AMD, in which abnormal blood vessels proliferate underneath the retina and macula, with resultant leakage, hemorrhage, and possible pigment epithelial detachment.

Relative to its dry counterpart, wet AMD has traditionally resulted in a greater proportion of patients with vision loss; however, the discovery of the association of vascular endothelial growth factor (VEGF) with its pathophysiology and subsequent development of targeted therapeutics has been a boon for its treatment. While anti-VEGF medications (e.g., aflibercept, brolucizumab, ranibizumab, etc.) remain imperfect and fail to demonstrate a clinical response in certain patients [8], their positive effects on long-term visual outcomes are well-established [9,10,11]. Unfortunately, treatment for dry AMD remains elusive, partially due to an incomplete understanding of the mechanisms involved in its pathogenesis.

Dysregulation in the complement cascade exhibits a particularly robust relationship with dry AMD [12,13,14], offering a strategy for treatments specific to the disease. Considering the constant advances occurring in our understanding of GA, the objective of this review is to provide a discussion of the broad mechanisms implicated in the progression of dry AMD while systematically focusing on the therapeutics previously and currently in development to target the complement cascade (Table 1).

## 2. Methods of Literature Search

To identify all therapeutics targeting components of the complement cascade, a comprehensive literature search was conducted using PubMed, Google Scholar, Web of Science, Clinicaltrials.gov, and company websites. The latter enabled the identification of pertinent findings presented at recent conferences but not yet published in the literature. Articles in the English language were included exclusively. This literature search was conducted up to the end of May 2022.

## 3. Risk Factors

Various modifiable and nonmodifiable demographic, environmental, and genetic factors have been recognized as predicting risk for GA, although their exact roles in disease progression remain uncertain [15].

### 3.1. Demographic and Environmental

Advanced age has been established as the primary contributor to GA development, with one analysis indicating prevalence rates of late stage AMD ranging from 0.1% among individuals aged 55–59 years to 9.8% in individuals aged 85 years and older [16]. The strongest modifiable risk factor for GA is smoking, with up to a four-fold increase in risk observed [17,18,19,20]. Other modifiable factors that have been explored include body mass index (BMI), diet, physical activity, and sunlight exposure [18,19,21,22,23,24,25,26]. Unfortunately, findings from multiple studies are conflicting, and there remains a need for further research to determine conclusive associations.

### 3.2. Genetic

Genomic investigations have profoundly expanded our understanding of the influence of specific genes in the development of AMD. In the original Genome-Wide Association Study (GWAS), 52 variants among 32 genetic loci were identified, which were predominantly involved in pathways related to the complement cascade, extracellular matrix remodeling, and lipid metabolism [27]. Findings from a more recent GWAS demonstrated enrichment among the complement, immune response, and protein regulation pathways across loci associated with disease progression [28].

The discovery of polymorphisms in two primary genetic loci has specifically established the genetic predisposition of individuals towards the development of GA secondary to dry AMD. The rs1061170 variant of the complement factor H (CFH) gene increases the incidence of early AMD and eventual progression to late stage AMD [14]. Similarly, the rs10490924 variant located near the age-related maculopathy susceptibility 2 (ARMS2) and high-temperature requirement A serine peptidase 1 (HTRA1) genes are strongly attributed to elevated risk for AMD and disease progression [29,30].

Variants in genetic loci within other regions are additional considerations when assessing risk. Particularly, gene polymorphisms related to the complement cascade (C2, C3, complement factor B [CFB], and complement factor I [CFI]) and lipid metabolism (apolipoprotein E [APOE], cholesterol ester transfer protein (CETP), and lipase C [LIPC]) modulate disease incidence and progression [15,20,31,32,33]. Summarily stated, these variants emphasize the role of local inflammatory processes in the pathogenesis of dry AMD and its progression to GA.

## 4. General Pathogenesis of Dry AMD

As evidenced by prior investigations, dry AMD is a multifactorial disease entity with various genetic and environmental elements contributing toward pathogenesis that remains entirely elusive. One component of this process is the overproduction of reactive oxygen species, which translates to oxidative stress, free radical formation, and peroxidation of the RPE [34,35,36]. Oxidative stress particularly obstructs the functionality of RPE stem cells by inhibiting mitochondrial activity [35,37,38,39], thereby creating a pro-inflammatory microenvironment unconducive to appropriate retinal development. Indeed, human RPE cells exposed to greater concentrations of oxidative stress demonstrate a dose-dependent reduction in viability [40]. Compounded with other factors, these oxidative insults result in the formation of drusen, which are foundational to the progression of dry AMD [41].

This formation and enlargement of drusen inhibits the ability of Bruch’s membrane (BM) to transport oxygen, nutrients, and metabolic substances by creating RPE detachments [42]. In turn, photoreceptor debris accumulates, resulting in further accretion of metabolites toxic to the RPE, such as lipofuscin and A2E [43,44]. Drusen additionally contain multiple pro-inflammatory markers, thus suggesting the importance of these processes towards the development of dry AMD [45]. These include components of the complement cascade, such as activators, fragments, and the membrane attack complex (MAC) [46]. The collective consequence of the aforementioned events is GA, a condition demarcated by outer retinal atrophic lesions that occur secondary to the loss or attenuation of the choriocapillaris, photoreceptors, and RPE [47,48].

## 5. Complement Cascade

The complement cascade is a component of the innate and adaptive immune systems, possessing a critical role in the defense against pathogens and the maintenance of homeostasis. Three principal pathways are involved in complement: the classical, alternative, and lectin pathways, as highlighted in Figure 1. Each is comprised of a series of reactions that ultimately conclude in the creation of a MAC [49]; nonetheless, the proteins involved in the initiation of each mechanism are distinct.

### 5.1. Pathways of the Complement Cascade

The classical pathway is initiated through the binding of antigen–antibody complexes to the C1q protein, the alternative pathway is initiated through a binding interaction between the C3b protein and foreign substances, and the lectin pathway involves the interaction of mannose-binding lectin with mannose-based polysaccharides on microbials. While the lectin and alternative pathways are effector arms of innate immunity, the classical pathway is an effector arm of adaptive immunity. Nevertheless, all three pathways converge at C3 convertase, leading to the cleavage of C3 into C3a and c3b, formation of C5 convertase, cleavage of C5 into C5a and c5b, and eventual formation of the MAC and subsequent cell destruction [49]. As a consequence of their fundamental roles, C3, C5, and the MAC have emerged as crucial targets for the therapeutic pipeline of dry AMD.

### 5.2. Complement Cascade in AMD

As alluded to previously, overactivity of the complement pathway has been heavily implicated in the pathogenesis of drusen [50]. Several proteins involved in the complement system have been identified as possible actors in the development and progression of dry AMD, including C3, C5, CFH, CFI, complement factor D (CFD), and complement factor B (CFB).

Evidence of this association is abundant in both experimental and clinical investigations. Among mice with inactivated C3, the rate of CNV was significantly lower following laser photocoagulation, thereby highlighting the protein’s role in the pathogenesis of AMD [51,52]. Other investigations have reported the presence of C5 in drusen and its elevation in the bloodstream of patients with AMD. C5a has been identified as having numerous inflammatory actions and thus is an effective target in reducing retinal loss in murine models [53,54]. MAC tends to accumulate in regions adjacent to microvascular injury, an early indicator of AMD; therefore, modulating MAC may represent a mechanism to prevent and/or limit the progression of GA [53,55,56]. Furthermore, downstream products of CFB have been observed to be elevated in AMD patients, inactivation of CFD has been linked to decreased photoreceptor loss, CFP stabilizes C3 and C5 convertases, and CD59 has been a basis for gene therapy specific to MAC [57,58,59].

## 6. Current Therapeutic Targets

Current management of dry AMD is centered around clinical observation and routine follow-up diagnostics and evaluations. Although there have been multiple clinical trials targeting components of the complement cascade for dry AMD (Table 1), an effective therapeutic is heretofore unavailable.

### 6.1. C1Q

#### ANX007

ANX007 (Annexon Inc., Brisbane, CA, USA) is a recombinant monoclonal antibody with an antigen-binding fragment that inhibits c1q. Through its action on c1q, the classical complement pathway is inhibited alongside C3 and C5 [60,61]. In a murine model of retinal photooxidative damage, ANX-M1, a monoclonal anti-C1q antibody similar to ANX007, reduced retinal atrophy [62]. Furthermore, a phase I trial (NCT04188015) using 2.5 mg and 5 mg of intravitreal (IVT) ANX007 demonstrated the safety and tolerability of both doses in 17 patients with primary open-angle glaucoma [60,61].

Because of these results, the phase II multicenter, randomized ARCHER study (NCT04656561), currently in its recruiting stage, will be conducted on an anticipated 240 patients with AMD to evaluate ANX007′s effect on GA growth rates. Patients will be randomized in a 2:2:1:1 ratio to receive ANX007 monthly, ANX007 every other month (EOM), sham monthly, or sham EOM, respectively.

### 6.2. C3

#### 6.2.1. AMY-106 (Cp40-KKK)

AMY-106, derived from Cp40-KKK, a fourth-generation compstatin analog, is a novel C3 inhibitor in development by Amyndas Pharmaceuticals [63,64]. This therapeutic maintained intraocular residence for more than 90 days after one 0.5 mg IVT injection in an investigation involving cynomolgus monkeys. Moreover, AMY-106 exhibited notable retinal tissue penetrance, as it localized with C3 in the choriocapillaris [63]. Provided its promise for dry AMD treatment, a phase I trial of AMY-106 is in development [64].

#### 6.2.2. Pegcetacoplan (APL-2)

Pegcetacoplan, additionally denoted as APL-2, is a PEGylated peptide inhibitor of C3 formulated by Apellis Pharmaceuticals that is administered intravitreally [65]. The agent was evaluated in the FILLY trial (NCT02503332) [65], a multicenter, randomized phase II trial encompassing 246 patients with GA. Subjects were randomized in a 2:2:1:1 ratio to injections of 15 mg pegcetacoplan monthly or EOM, or sham injections monthly or EOM for 12 months. Follow-up was conducted at 15 and 18 months. Pegcetacoplan treatment produced statistically significant reductions in GA growth rates. Compared to sham treatment, monthly pegcetacoplan treatment resulted in a 29% (95% confidence interval [CI], 9–49; *p* = 0.008) reduction in GA growth rate, while EOM treatment produced a 20% reduction (95% CI, 0–40; *p* = 0.067) [65].

Post hoc analysis suggested pegcetacoplan may be useful in AMD prior to the development of GA [66]. Incomplete RPE and outer retinal atrophy (iRORA) without neovascularization are considered precursor lesions to GA, while progression to complete RPE and outer retinal atrophy (cRORA) without neovascularization are synonymous with GA. In the trial, patient eyes receiving pegcetacoplan possessed lower rates of progression from iRORA to cRORA. As such, although findings did not achieve statistical significance, pegcetacoplan may prevent the incidence of GA [66]. However, in the FILLY study, new-onset wet AMD was more prevalent in the pegcetacoplan groups compared with the sham arm. Thus, participants who developed wet AMD with pegcetacoplan discontinued treatment [65].

An additional post hoc analysis of the FILLY trial determined 20.9% (18 of 86), 8.9% (7 of 79), and 1.2% (1 of 81) of study eyes developed wet AMD in the pegcetacoplan monthly, pegcetacoplan EOM, and sham groups, respectively, throughout the 18-month study course. Importantly, eyes that eventually developed wet AMD demonstrated increased baseline prevalence of both the double-layer sign (DLS), a finding suggestive of type I macular neovascularization (MNV), and wet AMD in the fellow eye [67]. Irrespective of the inclusion or exclusion of data for patients with new onset wet AMD, the study’s primary endpoint of reducing GA growth rate was still attained in the FILLY trial. Furthermore, an independent safety monitoring committee permitted completion of the study, as the exudations did not appreciably impact visual acuity [67].

Due to its acceptable safety profile and statistically significant reductions in GA lesion expansion, pegcetacoplan was cleared for multicenter, randomized phase III trials, labeled DERBY (NCT03525600) and OAKS (NCT03525613), both of which have completed enrollment (*n* = 621 and *n* = 638, respectively) [68], but remain ongoing [65,67]. During these investigations, patients who develop wet AMD in the study eyes will be concomitantly treated with anti-VEGF therapy, enabling subjects to continue receiving pegcetacoplan. This protocol will facilitate further delineation of the advantages and disadvantages of continued pegcetacoplan administration in addition to potential linkages between pegcetacoplan and new onset wet AMD [65,67]. Preliminary data at 18 months from the DERBY and OAKS trial have been encouraging. A combined analysis of the trials showed a 13% GA growth rate reduction in foveal lesions in the monthly (*p* = 0.0070) and EOM (*p* = 0.0069) pegcetacoplan groups relative to sham. Moreover, the GA growth rate of extra-foveal lesions was reduced by 21% (*p* = 0.0006) and 26% (*p* < 0.0001) in the EOM and monthly groups, respectively. Regarding new onset wet AMD in the combined studies, rates of 9.5% (40 of 419), 6.2% (26 of 420), and 2.9% (12 of 417) were noted at month 18 in the monthly, EOM, and sham arms, respectively. Across the pegcetacoplan monthly and EOM arms, 4 cases of endophthalmitis have been reported, translating to an infection rate of 0.044% per injection [69].

Another planned investigation is the GALE phase III trial (NCT04770545), a multicenter, non-randomized extension study that will further evaluate the safety and efficacy of pegcetacoplan using patients from the initial phase I trial (NCT03777332) and patients who complete 24 months of treatment from the DERBY and OAKS trials. Recruitment of 1200 patients is anticipated.

#### 6.2.3. POT-4 (AL-78898A)

POT-4 (AL-78898A) is a compstatin analog originally developed by Potentia Pharmaceuticals that functions as a C3 inhibitor and was the first complement inhibitor tested in humans with AMD [60,70]. Phase I trial (NCT00473928) results revealed no adverse drug events (ADEs) or serious adverse events (SAEs) with doses of up to 450 mg for patients with wet AMD [70]. Based on the tolerability of POT-4, a multicenter, randomized phase II trial (NCT01603043) with 10 patients was conducted to assess its efficacy in reducing GA lesion growth among individuals with dry AMD. However, the study was terminated prematurely, as four of seven participants (57.14%) in the POT-4 group developed drug product deposits in the eye.

#### 6.2.4. NGM621

NGM621 (NGM Biopharmaceuticals, San Francisco, CA, USA) is a humanized immunoglobulin G1 monoclonal antibody formulated as a C3 inhibitor. A phase I trial (NCT04014777) included 15 patients with GA secondary to dry AMD [71]. Participants were treated with either single-ascending doses (2 mg, 7.5 mg, 15 mg) of NGM621 or two doses of 15 mg delivered 4 weeks apart. Monitoring occurred for 12 weeks within all cohorts, and results indicated an appropriate safety profile, with no SAEs, ADEs, or new-onset CNV reported.

These promising findings resulted in the development of a multicenter, randomized phase II trial entitled CATALINA (NCT04465955) investigating the safety and efficacy of 15 mg of NGM621 administered every 4 or 8 weeks compared to sham throughout a 52-week interval [71]. The primary outcome is the rate of change in the GA lesion area over the trial period. In total, 320 participants with GA are currently participating in this ongoing trial.

### 6.3. C5

#### 6.3.1. Eculizumab

Eculizumab (Alexion Pharmaceuticals, Inc) is a humanized monoclonal antibody directed against C5 [72,73]. Systemic eculizumab is currently approved for the treatment of paroxysmal nocturnal hemoglobinuria and atypical hemolytic uremic syndrome. Thus, considering the evidence for the relationship of the complement cascade with AMD, the COMPLETE study (NCT00935883) was designed [72,73].

The single-center, randomized phase II trial evaluated the safety and efficacy of intravenous (IV) eculizumab for the treatment of GA among individuals with dry AMD. Overall, 60 patients were enrolled, with patients stratified into two distinct cohorts characterized by the presence of GA or the presence of drusen but the absence of GA (indicative of intermediate dry AMD). Each cohort of 30 patients was subsequently enrolled in a 2:1 ratio to IV eculizumab or placebo. The first 10 patients in the eculizumab group were provided with a low-dose regimen of 600 mg weekly for 4 weeks, followed by 900 mg every 2 weeks until week 24 of the study. The remaining 10 patients were provided with the high-dose regimen of 900 mg weekly for 4 weeks, followed by 1200 mg every 2 weeks until week 24. In the GA cohort, therapeutic efficacy was evaluated through a change in the square root of the lesion area at week 26. Although eculizumab demonstrated acceptable tolerability, average differences in the square root of GA area at 26 weeks were 0.19 (±0.12) mm and 0.18 (±0.15) mm in the eculizumab and placebo groups, respectively (*p* = 0.96) [72]. In the drusen cohort, therapeutic efficacy was evaluated through the presence of a 50% reduction in drusen volume at week 26. No patients treated with eculizumab achieved this primary outcome [73].

#### 6.3.2. Avacincaptad Pegol (Zimura^®^ [ARC1905])

Avacincaptad pegol (Zimura^®^), by Iveric Bio (Cranbury, NJ, USA), is a PEGylated RNA aptamer that operates as a C5 cleavage inhibitor, thereby impeding the complement cascade irrespective of the initial activation pathway. This agent was evaluated in the phase II/III GATHER1 trial (NCT02686658), a multicenter, randomized investigation. Randomization was conducted in two stages. In total, 77 part I participants were randomized in a 1:1:1 ratio to injections of 1 mg of avacincaptad pegol, 2 mg of avacincaptad pegol, and sham. Then, 209 part II participants were randomized in a 1:2:2 ratio to 2 mg of avacincaptad pegol, 4 mg of avacincaptad pegol, and sham. Relative to sham, subjects receiving 2 mg of avacincaptad pegol and 4 mg of avacincaptad pegol experienced a 27.4% (*p* = 0.0072) and 27.8% (*p* = 0.0051) reduction, respectively, in the mean rate of GA growth [74].

Avacincaptad pegol was generally well-tolerated, as no ADEs or SAEs were noted following 12 months of treatment. Compared to pegcetacoplan, a decreased incidence of endophthalmitis and CNV was reported with avacincaptad pegol [74]. Ongoing phase III trials will be beneficial to further exploring this phenomenon [65,74]. The 18-month results from the GATHER1 clinical trial continue to validate the safety and efficacy of avacincaptad pegol [75]. Compared to sham, the cohort treated with 2 mg of avacincaptad pegol demonstrated a 28% reduction (*p* < 0.0014) in GA growth and the cohort treated with 4 mg of avacincaptad pegol demonstrated a 30% reduction (*p* < 0.0021) from baseline to 18-month follow-up. No ADEs were reported up to 18 months post-administration with avacincaptad pegol [75].

Based on the efficacy and safety of avacincaptad pegol, another phase III trial, the multicenter, randomized GATHER2 study (NCT04435366), which compares 2 mg of avacincaptad pegol with sham, is currently being conducted. If data from the 448 patients currently enrolled in GATHER2 similarly reveal statistically significant reductions in GA growth rate with avacincaptad pegol, the treatment will have the two phase III trials necessary for FDA approval application [74].

#### 6.3.3. Tesidolumab (LFG316)

Tesidolumab (LFG316), a monoclonal C5 inhibitor developed by Novartis Pharmaceuticals (Basel, Switzerland), was investigated in a multicenter, randomized phase II clinical trial (NCT01527500). In the study, 158 patients were enrolled and separated into two groups. Part A examined the safety and efficacy of multiple 5 mg IVT injections of LFG316 relative to sham every 28 days over the course of 505 days. Part B examined the safety and pharmacokinetic properties of a single 10 mg IVT injection. Results revealed a relatively benign safety profile, with no demonstrable improvement in the primary outcome of GA lesion growth or the secondary outcome of BCVA.

### 6.4. Complement Factor B

#### IONIS-FB-lrx

In more recent years, therapies have emerged to alter gene expression rather than target proteins. IONIS-FB-lrx (Ionis Pharmaceuticals, Carlsbad, CA, USA) is a novel anti-sense oligonucleotide (ASO) targeting the gene encoding complement factor B (CFB), a moiety of the alternative complement pathway. When delivered subcutaneously, IONIS-FB-lrx reduced circulating levels of CFB in a dose-dependent manner among 54 healthy participants of a phase I trial [76]. No notable adverse effects were reported. Given these results, the multicenter, randomized phase II GOLDEN trial (NCT03815825) has been initiated to evaluate the influence of IONIS-FB-lrx administration on the progression of GA lesion size [76]. Anticipated recruitment is 330 patients.

### 6.5. Complement Factor D

#### Lampalizumab (FCFD4514S)

Complement factor D (CFD), a rate-limiting enzyme of the alternative pathway, converts proconvertases into active C3 and C5 convertases. Consequently, inhibition of CFD has surfaced as an attractive therapeutic option for dry AMD [77]. Lampalizumab, designed by Genentech (San Francisco, CA, USA), is a humanized monoclonal antibody with an antigen-binding fragment that inhibits complement factor D.

In a phase I trial (NCT00973011), 18 participants received 10 mg of IVT lampalizumab. The agent had an acceptable safety profile without notable ocular or systemic ADEs or SAEs. To expand these findings, the MAHALO phase II trial (NCT01229215) was conducted to evaluate lampalizumab’s efficacy in reducing GA progression [78]. Monthly treatment with lampalizumab resulted in a statistically significant 20% reduction (*p* = 0.117) in mean GA lesion enlargement relative to sham. Patients with high-risk complement factor I (CFI) alleles displayed a greater response to monthly lampalizumab with a 44% reduction (*p* = 0.0037) in GA atrophy progression compared to sham. Similar to phase I findings, lampalizumab was well-tolerated [78].

These findings encouraged the design of the CHROMA (NCT02247479) and SPECTRI (NCT02247531) multicenter, randomized phase III trials to further delineate lampalizumab’s effects on GA enlargement [79]. Both investigations randomized 1881 total participants in a 2:1:2:1 configuration to 10 mg of IVT lampalizumab every 4 weeks, sham every 4 weeks, 10 mg lampalizumab every 6 weeks, or sham every 6 weeks, over a period of 96 weeks. Efficacy was assessed as a change in GA area from baseline to week 48 of the study. Despite acceptable tolerability, lampalizumab was determined not to be efficacious for the treatment of GA. 48 weeks post-administration, the difference in means (lampalizumab vs. Sham pooled) was 0.071 mm^2^ (*p* = 0.25) for lampalizumab every 4 weeks and 0.070 mm^2^ (*p* = 0.25) for lampalizumab every 6 weeks. In contrast to the MAHALO study, both CHROMA and SPECTRI identified no relationship between CFI risk alleles and GA progression. Random chance and small sample size in the MAHALO study were posited to have contributed to this discrepancy [79].

Visual function decline among patients in the CHROMA and SPECTRI trials was examined utilizing several metrics, including best corrected visual acuity (BCVA), low-luminance visual acuity (LLVA), maximum reading speed, microperimetry, and patient-reported outcomes (PRO), as quantified via the Functional Reading Independence (FRI) Index and the National Eye Institute Visual Function Questionnaire (NEI VFQ-25). Results of the analysis revealed no statistically significant difference in visual function assessments across all treatment groups at week 48 of the phase III trials [80].

### 6.6. Complement Factor H

#### 6.6.1. AdCAGfH

CFH downregulates the alternative complement cascade by preventing C3 convertase formation from c3b; therefore, it has been postulated as a potential therapeutic for dry AMD. The injection of AdCAGfH, an adeno-associated virus (AAV) containing murine factor H, demonstrated an attenuation of C3-induced retinal inflammation [81]. This gene therapy is currently under consideration for human subjects [82].

#### 6.6.2. GEM103

GEM103 is a recombinant human CFH developed by Gemini Therapeutics (Cambridge, MA, USA). In a phase I clinical trial (NCT04246866) comprised of 12 participants, GEM103 demonstrated minimal adverse effects and a dose-dependent increase in levels of CFH [83]. Significantly, no CNV was observed, a phenomenon previously reported in the literature in association with repeated administration of IVT complement inhibitors [67].

Interim findings from the subsequent multicenter, randomized ReGAtta phase IIa study (NCT04643886) have suggested positive alterations in biomarkers linked to the complement-mediated pathogenesis of dry AMD. The safety profile of GEM103 remained relatively benign, with adverse events reported among 16 of 62 volunteers, of which one case of mild iritis was directly related to the compound [84]. Since the ReGAtta trial achieved its primary endpoint of evaluating the safety and tolerability of GEM103, the trial was terminated [85].

### 6.7. Complement Factor I

#### GT005

CFI is a regulator of the alternative pathway. With the assistance of its cofactors, it cleaves C3b into pro-inflammatory ic3b, which is subsequently cleaved into inert c3dg. Thus, the protein is crucial for modulating the activation of the alternative pathway. Importantly, experimental investigations confirmed the ability of AAV-containing CFI cDNA to induce CFI protein expression in human RPE lines and murine retina, thereby stimulating the development of GT005 by Gyroscope Therapeutics.

The multicenter, randomized FOCUS phase I/II trial (NCT03846193), which is currently recruiting patients (65 anticipated), is evaluating this agent among individuals with GA secondary to dry AMD [86]. Interim data for 31 patients demonstrated a significant increase in vitreous CFI and a significant reduction in C3 and its cleavage products following administration of GT005. No SAEs were reported in the treatment cohort [87].

Additional multicenter, randomized phase II trials, entitled HORIZON (NCT04566445) and EXPLORE (NCT04437368) are evaluating a two-dose single injection of GT005 in patents with GA from dry AMD. The HORIZON and EXPLORE trials will include patients previously genotyped for rare CFI genetic variants, enabling elucidation of the interaction between the therapeutic’s effects and the presence of CFI mutations on the advancement of GA. Anticipated recruitment is currently 250 and 75 patients, respectively.

### 6.8. Membrane Attack Complex (MAC)

#### AAVCAGsCD59 (HMR59)

Regulators of the complement cascade are emerging targets for investigational compounds. AAVCAGsCD59, also denoted HMR59, is an intravitreally injected AAV vector developed by Hemera Biosciences that induces the formation of CD59, thereby preventing the binding of C9 required for the formation of a complete MAC [88]. Pre-clinically, HMR59 demonstrated efficacy in attenuating laser-induced CNV among murine models [89].

The interim results from a non-randomized phase I trial involving 17 eyes with dry AMD (NCT03144999) have indicated promise, with acceptable tolerability and no dose-limiting toxicity. While adverse events were limited, it is important to note that four eyes developed mild inflammation that subsequently resolved with topical steroid therapy or observation. Two of these four patients additionally required topical medication to reduce intraocular pressure. Encouragingly, although not powered to illustrate efficacy, findings from the investigation revealed that 9 of 11 patients had decreased GA progression relative to historical controls [90]. As such, a multicenter, randomized phase II trial is currently planned (NCT04358471) to evaluate the therapeutic’s effect on the expansion of the GA area. 

### 6.9. Properdin

#### CLG561

Properdin is a positive regulator of the alternative complement pathway, preventing the degradation of C3 and C5 convertases. Thus, CLG561, an anti-properdin antibody, was developed as a potential therapeutic by Alcon Research [91]. Results from a phase I trial (NCT01835015) showed IVT doses of CLG561 up to 10 mg were safe and well-tolerated, with no systemic or ocular ADEs reported [92].

Subsequently, a multicenter, randomized phase II trial (NCT02515942) was conducted to evaluate CLG561, both individually and in combination with LFG316 (tesidolumab), against sham in 114 participants with GA. Twelve total injections were administered per group, with injections occurring every 28 days. Change in GA lesion size from baseline to day 337 was the primary outcome. Neither CLG561 monotherapy (*p* = 0.1019) nor CLG561 plus LFG316 (*p* = 0.5987) significantly reduced GA lesion growth relative to sham at day 337.

## 7. Challenges and Future Directions

Translation of experimental research for dry AMD is partially impeded by experimental models that are incapable of elucidating and replicating the complex pathogenic processes in human eyes [36,93]. As the disease is a multifactorial process, it is essential for these models to incorporate variables such as oxidative stress, free radicals, and peroxidation to ensure that results are applicable to a clinical environment, a process that has hitherto remained elusive.

In recent years, several experimental animal models have been established to identify characteristics of dry AMD similar to the pathology observed in humans. Traditionally, rodents have been preferred to primates because of their short life cycle, rapid disease progression, and genetic similarity to humans. One challenge associated with the rodent model is the crucial differences in the anatomical structures of the eye. While dry AMD primarily affects central vision through degeneration of the fovea and macula, these structures are absent in rodents [36,94,95]. Apart from anatomy, other key AMD pathology such as drusen, RPE atrophy, and CNV are not exhibited in rodent models [95,96]. These differences preclude a comprehensive understanding of the mechanism of AMD, thereby hindering the discovery of novel therapeutics. Advancements in this arena are currently being propelled by the development of three-dimensional in vitro models using human cells to represent the pathology entwined within BM and the RPE layer [97]. Such an innovation would provide essential insights into the biomechanical performance of the retina and replace current in vivo animal models that provide an incomplete understanding of the disease.

Nevertheless, even where experimental research may be translated into the development of clinical compounds, modulating the complement cascade creates challenges for a myriad of reasons. Interference in this immunogenic process may result in innumerable adverse effects, an essential consideration for therapeutics administered over an extended period for the management of chronic diseases. To ameliorate this concern, pharmacologic inhibition should exclusively target the dysregulated components of the complement cascade, thus limiting the associated systemic effects. However, extensive exploration remains to delineate the mechanisms of disease progression in order to successfully produce such compounds.

Furthermore, there is notable intrapopulation variation in the expression of genetic risk factors contributory to the pathogenesis of dry AMD [15,20,46,98,99]. Because complement dysregulation comprises merely a single component of this multifactorial condition, the selection of individuals possessing a particular predisposition to complement-mediated disease progression would enable the development of personalized treatments for patients who would extract the most benefit. Identifying specific genetic variants associated with increased risk (e.g., rs1061170 variant of CFH) or calculating aggregate genetic risk profiles (from multiple variants) to stratify patients are potential approaches to facilitating more efficacious clinical trials. Alternatively, investigational compounds targeting pathways implicated in extracellular matrix remodeling, lipid metabolism, or oxidative stress would be of greater suitability for individuals not possessing these critical genetic risk factors [100,101,102,103].

Clinically, patient adherence to a structured treatment regimen is a substantial challenge that persists across the ophthalmic landscape [104,105]. To circumvent this barrier, innovative modalities of ocular delivery such as biodegradable polymeric implants, colloids, and hydrogels are currently being explored as potential solutions [106,107,108,109]. These sophisticated alternatives have the capacity to rapidly accelerate the adoption of dry AMD therapeutics.

## 8. Conclusions

Despite the absence of available therapeutics for patients with GA secondary to dry AMD, there remains significant hope that one of the numerous clinical trials will produce an effective treatment option. The probable association of the complement cascade with the pathogenesis of dry AMD, as evidenced by genetic, experimental, and clinical investigations, emphasizes the integral participation of inflammation in the development and progression of the disease. This pathway offers a prolific target for the synthesis of novel therapeutics. As a continued exploration of the linkages between the complement cascade and GA secondary to dry AMD occurs, our understanding of the disease will expand beyond its current state, enabling further identification of treatments for patients.

## Figures and Tables

**Figure 1 biomedicines-10-01884-f001:**
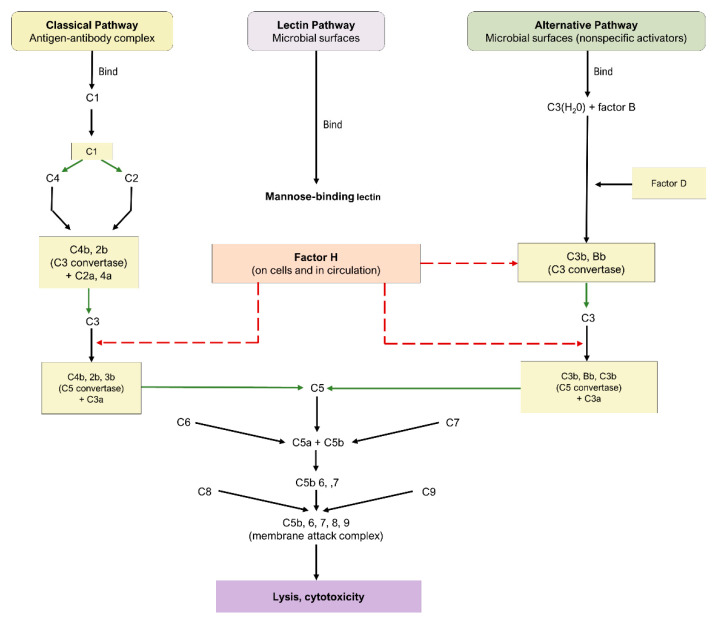
Complement cascade pathway.

**Table 1 biomedicines-10-01884-t001:** Completed and current investigational treatments targeting the complement cascade in dry age-related macular degeneration.

Therapeutic Category or Name	Mechanism	Mode of Administration	Clinical Trial ID, NCT# *	Study Phase (Status) ^†^	Patients, *n*	Primary Outcome
**C1Q** ANX007						
	Recombinant monoclonal antibody	Intravitreal	NCT04656561	Phase II (Recruiting)	240 (Anticipated)	GA growth
**C3**			N/A NCT03525600 NCT03525613 NCT04770545 NCT01603043 NCT04465955	N/A Phase III (Ongoing) Phase III (Ongoing) Phase III (Recruiting) Phase II (Terminated, Adverse effects) Phase II (Ongoing)		
AMY-106 (Cp40-KKK)	Compstatin analog inhibitor	Intravitreal			N/A	N/A
Pegcetacoplan (APL-2)	PEGylated peptide inhibitor	Intravitreal			621 638 1200 (Anticipated)	GA growth GA growth Safety
POT-4 (AL-78898A)	Compstatin analog inhibitor	Intravitreal			10	GA growth
NGM621	Humanized monoclonal antibody	Intravitreal			320	GA growth
**C5** Eculizumab Avacincaptad Pegol (Zimura^®^ [ARC1905]) Tesidolumab (LFG316)	Humanized monoclonal antibody PEGylated RNA aptamer Human monoclonal antibody	Intravenous Intravitreal Intravitreal	NCT00935883 NCT04435366 NCT01527500	Phase II (Completed, Ineffective) Phase III (Ongoing) Phase II (Completed, Ineffective)	60 448 158	GA growth/Reduction of drusen volume GA growth GA growth
**Complement Factor B** IONIS-FB-LRx	Anti-sense oligonucleotide	Subcutaneous	NCT03815825	Phase II (Recruiting)	330 (Anticipated)	GA growth
**Complement Factor D** Lampalizumab (FCFD4514S)						
	Humanized monoclonal antibody	Intravitreal	NCT02247479 NCT02247531	Phase III (Terminated, Ineffective) Phase III (Terminated, Ineffective)	906 975	GA growth GA growth
**Complement Factor H** AdCAGfH GEM103						
	Adeno-associated virus	Subretinal gene therapy	N/A	N/A	N/A	N/A
	Recombinant human CFH	Intravitreal	NCT04643886	Phase IIa (Terminated, Effective)	62	Safety
**Complement Factor I** GT005						
	Adeno-associated virus	Subretinal gene therapy	NCT03846193 NCT04566445 NCT04437368	Phase I/II (Recruiting) Phase II (Recruiting) Phase II (Recruiting)	65 (Anticipated) 250 (Anticipated) 75 (Anticipated)	Safety GA growth GA growth
**Membrane Attack Complex (MAC)** AAVCAGsCD59 (HMR59)						
	Adeno-associated virus	Intravireal gene therapy	NCT04358471	Phase II (Hiatus)	N/A	GA growth
**Properdin** CLG561						
	Human antibody Fab	Intravitreal	NCT02515942	Phase II (Completed, Ineffective)	114	Safety/IOP change/GA growth

Abbreviations: NCT, National Clinical Trial; GA, geographic atrophy; IOP, intraocular pressure. * NCT# was unavailable for some therapeutics as formal clinical trials have yet not been conducted; NCT# is provided for the most recent clinical trials. ^†^ Where applicable, the most recent study phase is listed.

## Data Availability

No new data were created or analyzed in this study. Data sharing is not applicable to this article.

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
