# Peer review of "Targeting the Complement Cascade for Treatment of Dry Age-Related Macular Degeneration"

_biomedicines, 2022, doi:10.3390/biomedicines10081884_

Round 1

Reviewer 1 Report

In this review, the authors try to summarize the role of the complement system in dry AMD and discuss the emerging therapies in early-phase clinical trials. They conducted comprehensive literature search to include therapeutics targeting components of the complement cascade. The efforts for finding and testing the association between development and progression of dry AMD and key proteins in complement cascade are important and necessary. But there are several deficiencies in this study.

 1.      There were already several reviews about complement cascade and AMD published these years (Such as PMID: 34202223, PMID: 34999723). The authors should think more about the novelty and significance of their review.

2.      In the title and main text, the author mentioned “Dry Age-Related Macular Degeneration and Geographic Atrophy” multiple times. As we know, Geographic Atrophy is a special type of Dry AMD. The authors should pay attention to the expressions.

3.      In main text, 3. Risk Factors and 4. General Pathogenesis of AMD were not directly related to the topic. Pathogenesis section described development of general AMD, not even of dry AMD or GA.

4.      Table 1 should contain more information about the trails, such as number of patients, drugs administration, status and primary outcomes, etc.

5.      In 7. Challenges and Future Directions section, the authors only focus on experimental animal models. It is better to discuss the current challenges and future directions more comprehensive based on summary of previous content.

Reviewer 2 Report

The topic of this manuscript is actual and of interest for the ophthlmologic community.

I would suggest to include a table with the type of treatment (e.g. adenovirus, gene terapy, etc) and if the trial is ongoing or stopped

Moreover, a recent molecule inhibiting the CD59 is not mentioned. Probably it would be good to include it as well.

Round 2

Reviewer 1 Report

In their revised manuscript, the authors have adequately responded to the comments provided in the original review’s. They also included more discussions for helping comprehensively explain the current challenges and future directions based on the content. This review provides an updated summary of the growing armamentarium against the complement cascade in dry AMD. It will be a good contribution to the field.